# OpenReview forum: "SpatialVLA-Mamba: Efficient State-Space Models with Self-Refinement for Spatially-Grounded Robotic Control"
_ICLR.cc/2026/Conference — ICLR 2026 Conference Withdrawn Submission_

### Official Review · Reviewer_mKop · 2025-10-20

**Soundness:** 3
**Presentation:** 1
**Contribution:** 2
**Rating:** 2
**Confidence:** 3

**Summary:**

The work proposes a new VLA pipeline with a focus on 3 aspects, namely enhancing perception with metric aware structure, incorporating long horizon modeling via the use of a Mamba SSM policy, and a chain of thought reinforcement learning loop for self refinement. Experiments run in simulation show improved performance and robustness on tabletop manipulation tasks, and ablation suggest all three modifications play an important role.

**Strengths:**

The authors show strong results, with their proposed pipeline proving efficient on the Webots benchmark considered.

The work that has gone into building such a model with the multiple moving parts is in itself an engineering feat which can be appreciated.

Indeed the work is wide reaching with incorporations of modules from computer vision, state space models, and chain of thought reinforcement learning, which all have improvement to offer to robotics and especially VLAs which are the hot architecture of the moment.

**Weaknesses:**

**Lack of clear contribution**
- The authors claim in the intro (lines 87-95) that the contributions of the paper are the design of the encoder, swapping transformer policy with mamba, and the CoT-RL loop. None of these is a contribution taken on its own.
- Taking traditionally RGB only (or with depth) encoding approaches and adding a layer of bounding box detection finer grained information is not a real contribution especially when the stitching is done via pretrained models.
- Swapping an architecture with another to reap the known advantages of long term sequence handling and inference efficiency it offers is not a contribution (Mamba vs transformers)
- The CoT-RL pipeline appears to have required the most design decisions among the 3, yet it appears to be quite hacky with the summarizer used not really well explained and no Appendix to delve into the details. Perhaps there is a real contribution there but the current state of the manuscript fails to show how different the approach is from cited works (lines 234-235), or to clearly justify the overall structure.



**Lack of discussion on computational aspects**
- The model consists of multiple moving parts, most of which are frozen pre-trained modules for bounding box extraction, language encoding, image (RGB and Depth) encoding, LLM summarizer and decoder, Clip scorer, etc (frankly a bit of a Frankenstein's monster). There is no discussion on what the effective model size is and how much of it is trainable, at what cost this occurs compared to some of the other VLA models considered.
- The comparison in Table 3 is basically just comparing Attention to SSMs which has been extensively studied and offers no effectively novel insights here.

**Poor presentation**
- There is no appendix although a lot of information warrants a deep dive, given the paper is essentially stitching together three module modifications with a lot of moving parts and pretrained modules.
- Table 2 and Figure 3 are essentially the same thing, their combination occupies more than half a page to effectively report 6 numbers.
- The LLM use statement is duplicated in the ethics statement as well as the LLM use declaration

**Comparing apples to apples?**
- There is no presentation of the other models compared against, explanation of why this is a fair comparison (aren't most of them image only directly encoded ?). Also no explanation of omission of very popular VLAs such as the pi_0 family of models.

**Lack of real-world deployment**
- This is a robotics (purely) application paper. It is standard to expect real world transfer of the pipeline on hardware. This can constitute a lot off additional work but is crucial to showcase how well this can be applied beyond simulation benchmarks. (again given the authors can also circle back to their claims on robustness and unseen scenes)
- One would also highlight that if real world experiments should be omitted, a larger set of benchmarks should be established instead of a single experimental simulation framework (Webots). Is this a standard benchmark?

**Questions:**

- Why does only swapping Mamba with Transformers incur a 4% drop in performance in unseen tasks? Are these tasks all long horizon to account for this?

- Are long horizon tasks dealt with in one go, i.e. single text instruction fed in from the start? In this case is the policy just trying to imitate some long horizon sequence of movements or is there some understanding of atomic task completion? Is this not a caveat for needing long horizon modeling as most VLA with some reasoning capability would in practice have a planner to break things down into atomic actions to take in a sequence?

---

### Official Review · Reviewer_PtBv · 2025-10-24

**Soundness:** 2
**Presentation:** 2
**Contribution:** 2
**Rating:** 2
**Confidence:** 3

**Summary:**

The paper builds a VLA that explicitly encodes geometry (depth + simple spatial tokens), and replace the transformers decoder with a Mamba state-space model for linear-time long-horizon action generation, and on top of this adds a lightweight CoT RL loop that lets the agent summarize its own candidate plans and score them with CLIPscore.

**Strengths:**

1. The paper tries to address the three gaps: explicit spatial grounding (depth + geometric tokens), efficient long‑horizon decoding (Mamba SSM), and robustness via a self‑refinement loop.

2. The model outperforms other compared baselines in experiments.

3. The CoT‑RL loop uses internal summarization and CLIPScore rather than relying on a big external LLM, which is a practical design choice

**Weaknesses:**

1. There is no real robot experiments to verified the effectiveness of the proposed approach.

2. There isn't comparison with models that leverage action experts or speculative decoding or token pruning to improve test-time speed.

3. There isn't comparison to VLA model that leverage CoT such as MolmoAct, ECoT, or VLA-COT

**Questions:**

Questions are same to the weakness section.

---

### Official Review · Reviewer_324F · 2025-11-01

**Soundness:** 1
**Presentation:** 1
**Contribution:** 1
**Rating:** 0
**Confidence:** 5

**Summary:**

The paper introduces Spatial-Mamba, a Mamba-based state-space decoder VLA with designed spatial-aware encoder to improve the accuracy and robustness of VLA methods. The propose spatial-aware encoder integrates depth and object localization information into input multimodal embeddings and a CoT-RL pipeline is proposed to used CLIPScore as reward to achieve VLA policy self-refinement. The experiments are conducted on Webots simulations and various results are reported .

**Strengths:**

The authors introduce a dedicated Mamba-based based SLAM framework with sophisticated spatial-aware encoder and CoT-RL designs to accuracy and robustness of VLA various scenes. The designs seems reasonable and simulation experimental results are reported.

**Weaknesses:**

1.The novelty and results of this paper are limited. Replacing transformer with Mamba is common and have be exploited in RoBoMamba, the results of this paper do not show the superiority of the proposed method.

2.Some detail designs are not sound and unclear. Simple project all detected bbox and relative pose into multimodal input embedding is doubtful. The reward in Eq.10 is not reasonable, simple action label like move left, pick or place, are not suitable to compute with  task instruction.

3.The writing of technical details is unclear. The input and output of some components are unclear and the whole pipeline is confusing.

4.The experiments results is not sufficient and reliable. The experiments are only conducted on Webots simulation which is not the main benchmarks in VLA community. The PaLM-E method is not open-sourced, how the authors obtained the results of it. The comparison with current open source VLA (openvla, spatialvla, pi, gr00t) is missing.

**Questions:**

1.What is the shape of E_B, P, and E_P in Eq.3. Is the self-attention is used in Eq.4? what is the shape of H in Eq.4?

2.How exactly the multimodal dadptation is implemented, how the action vector a_t is calculated.

3.How the decoder in Eq.8 is trained, what data is used, how is the performance. And the authors should provide results on usefulness of reward in Eq.10.

4.How the candidate action is obtained from pretrained policy based on Mamba?

---

### Official Review · Reviewer_nhHi · 2025-11-06

**Soundness:** 3
**Presentation:** 2
**Contribution:** 2
**Rating:** 2
**Confidence:** 4

**Summary:**

The paper introduces SpatialVLA-Mamba, a vision-language-action (VLA) model aimed at improving spatial reasoning and long-horizon capability in robot manipulation. The problem setting follows the typical VLA formulation but takes depth image and geometric primitives as input: given an RGB-D visual observation, a language instruction, object bounding boxes and relative poses, the model outputs a sequence of low-level actions to complete the task. The motivation is that existing transformer-based VLAs often suffer from poor spatial grounding and have high computational costs when modeling long action sequences.

To address these issues, the authors propose three main components.
1. Spatial-aware encoder: the RGB-D images are processed by ViT, the detected object bounding boxes and relative poses are embedded by MLPs. Then these visual and geometry tokens are fused with text with a frozen T5 model.
2. Mamba-based decoder: instead of a standard transformer, the paper employs a state-space model (SSM) architecture based on Mamba, claiming linear-time complexity and better scalability to long horizons. The decoder processes the sequence of encoded visual-language features to predict the next action autoregressively.
3. Chain-of-Thought reinforcement learning: after supervised pretraining, the model undergoes a refinement stage. It generates textual outcome descriptions of its own action sequences, evaluates them with CLIPScore as an intrinsic reward, and fine-tunes the policy using PPO. This is framed as a self-evaluation loop that improves spatial grounding and task success without additional human labels.

The authors evaluate SpatialVLA-Mamba in Webots simulation using a 7-DoF robot arm for tabletop manipulation tasks. They report that their method reduces spatial localization errors and achieves higher success rates than baselines methods. Ablation studies suggest that both the spatial encoder and Mamba backbone contribute to performance gains, and the CoT-RL phase brings additional improvement. Efficiency measurements indicate reduced inference latency and memory usage compared with transformer decoders of similar scale.

**Strengths:**

- The motivation and problem formulation is sound: The paper explicitly identifies three major pain points in existing VLA systems: poor spatial reasoning, inefficiency on long-horizon tasks, and weak self-evaluation mechanisms. The authors map each to a proposed module. This clarity makes the overall system architecture easy to follow and motivates why each component exists.
- The proposed architecture (Spatial Encoder + Mamba Decoder + CoT-RL loop) is conceptually simple yet implementable within existing VLA training pipelines.
- The proposed method shows empirical improvement over existing VLA models: the combination of the three main modules yield consistent performance gains across simulated manipulation benchmarks, achieving about 35% lower spatial localization error and higher success rates.
- Solid ablation and component analysis: The experiments include clear ablations removing each of the three modules. The results show consistent performance drops when any component is omitted, which supports the claimed contributions.

**Weaknesses:**

- The authors didn't compare with some of the more popular recent VLA models such as OpenVLA, Pi-0 and Gr00t. Also, SpatialVLA emphasize spatial representations in recent VLA research.
- Limited evaluation details and unclear task scope. The authors did not provide sufficient information about the evaluation tasks: e.g., how many tasks, their diversity, how generalizable. This makes it hard to judge the significance of the results, especially the spatial error aspect.
- Simulation-only evaluation (and depth reliance) reduces real-world impact. The baselines that this paper compare to are all demonstrated on real hardware. Specifically, the RT-1 trajectories are collected on real robots, introducing a significant domain gap to the simulated environment.

**Questions:**

- The CLIPort model predicts pick and place poses instead of raw robot actions. How is it comparable with proposed method? Are the simulation tasks all pick-and-place style only?
- Does the CoT-RL phase require stepping the simulator? Section 3.4 mentions simulated episodes but from section 3.3 it appears that the reward depends solely on the action sequence.
- During inference, is the action sequence carried out in an open loop?

---

### Note · Authors · 2026-01-21

I have read and agree with the venue's withdrawal policy on behalf of myself and my co-authors.